# The Influence of Severity and Disease Duration on TNF Receptors’ Redistribution in Asthma and Rheumatoid Arthritis

**DOI:** 10.3390/cells12010005

**Published:** 2022-12-20

**Authors:** Alina Alshevskaya, Julia Lopatnikova, Julia Zhukova, Oksana Chumasova, Nadezhda Shkaruba, Alexey Sizikov, Irina Evsegneeva, Daria Demina, Vera Nepomniashchikch, Aleksander Karaulov, Sergey Sennikov

**Affiliations:** 1Federal State Budgetary Scientific Institution, “Research Institute of Fundamental and Clinical Immunology” (RIFCI), 630099 Novosibirsk, Russia; 2Federal State Autonomous Educational Institution of Higher Education I.M. Sechenov, First Moscow State Medical University of the Ministry of Health of the Russian Federation, 119435 Moscow, Russia

**Keywords:** TNF-alpha, bronchial asthma, rheumatoid arthritis, cellular immunology, cytokine receptors, immune regulation

## Abstract

One of the mechanisms of cellular dysfunction during the chronization of immune-system-mediated inflammatory diseases is a change in the profile of expression and co-expression of receptors on cells. The aim of this study was to compare patterns of redistribution of TNF receptors (TNFRs) among patients with different durations of rheumatoid arthritis (RA) or asthma. Subgroup analysis was performed on RA (*n* = 41) and asthma (*n* = 22) patients with disease duration<10 years and >10 years and on 30 comparable healthy individuals. The co-expression profile of TNFR1 and TNFR2 was assessed in T cells, B cells, monocytes, regulatory T cells, T-helper subsets, and cytotoxic T-lymphocyte subsets. Percentages of cells with different co-expression combinations and receptor density per cell were estimated. Longer disease duration was significantly associated with a redistribution of receptors in immunocompetent cell subsets with an increase in the expression of TNFR1 in asthma but did not correlate with significant unidirectional changes in receptor expression in RA. In asthma, a higher proportion of cells with a certain type of TNF receptor (as compared with the healthy group) was correlated with a simultaneous greater density of this receptor type. In RA, an inverse correlation was observed (compensatory lower receptor density). Mechanisms of long-term changes in the expression of TNF receptors differ significantly between the diseases of autoimmune and allergic etiology. The formation of irreversible morphostructural alterations was strongly correlated with changes in the expression of TNFR1 in asthma and with changes in the expression of TNFR2 in RA.

## 1. Introduction

As a pleiotropic cytokine with multiple regulatory and proinflammatory effects, TNF is involved in the pathogenesis of various disorders such as autoimmune, cardiovascular, and allergic diseases, cancers, and many other illnesses [1]. Signaling pathways can be activated by TNF through binding to two types of specific receptors (TNFR1 and TNFR2), which trigger, with different probabilities, two types of biological response to this mediator [2]. TNF performs its biological functions via two main signaling cascades: a nuclear factor kappa-light-chain-enhancer of activated B cells (NF-κB) pathway and a mitogen-activated protein kinase (MAPK) pathway [3,4]. These two types of signaling cascades can mediate either proliferation and survival or apoptosis and cell death, depending on which receptor binds to TNF and under what conditions [5,6,7].

Studies have shown that the regulation of a functional response of the cell to a cytokine may depend on the number of cell-surface receptors and on the ratio of different types of receptors [8], and these patterns can lead to a shift in the balance between proapoptotic and proliferative signaling pathways [4,9]. In this context, an important two-way relation is observed. On the one hand, a change in the balance of expression and co-expression of receptors (including their soluble forms) can cause the activation or an alteration of activity of pathological processes [10]. On the other hand, the initiated diseases will also affect levels of expression of different types of receptors, and this phenomenon may be one of the mechanisms regulating the activity of pathological processes [11].

Different actions of TNF and its receptors have been documented for allergic–atopic and autoimmune diseases. For instance, it has been demonstrated that in the pathogenesis of asthma, an impairment of TNF–TNFR signaling contributes to the polarization toward T helper 2 (Th2) and Th17 phenotypes, raises concentrations of inflammatory cytokines (IL-4, IL-5, IL-17, and TNF) in serum and bronchoalveolar lavage fluid, and exacerbates allergic airway inflammation [11]. By contrast, in the pathogenesis of rheumatoid arthritis (RA), a disturbance of signaling through TNF receptors creates an imbalance among cells of the immune system, enhances the production of autoantibodies, and increases the levels of proinflammatory cytokines [11].

Disease duration is an important determinant of the development of irreversible pathological changes in the human body. Research on RA patients has uncovered a correlation between disease duration and overall disease severity, resistance to standard therapy, and the progression of structural bone changes [11,12,13,14]. At the same time, a significant cutoff was identified in disease durations of 10 years, after which the frequency of complications goes up significantly [11,13]. Similar data are also available about patients with asthma, where the disease duration of 10 years is also considered a cutoff that is associated with significant irreversible impairment of external respiration function and with resistance to ongoing pathogenesis-targeting therapy [15,16,17]. Thus, in both types of these immune-system-mediated diseases, investigators highlight the disease duration of 10 years as a cutoff for the initiation of morphological and functional changes in the patient’s body.

Another important factor that exerts a significant effect on the severity of diseases, according to many studies, on the rate of their progression, and on the response to therapy is the parameters of the cytokine profile and of the receptors located on immunocompetent cells [4,7,18,19,20]. Different types of diseases (for example, atopic and autoimmune) have dissimilar, unique profiles of co-expression and of density of cytokine receptors on cells, and this profile contributes substantially to pathogenetic characteristics of the initiation and course of diseases [2,19,21]. Nonetheless, specific mechanisms of alterations in the cytokine–receptor interaction that cause these differences remain poorly understood. In addition, there is still an open question about temporal features of the changes seen in the pathogenesis of diseases and accordingly about opportunities for an early targeted therapeutic intervention.

Data on the differences in expression and co-expression profiles of type 1 and 2 TNF receptors in key immunocompetent cell populations in RA patients or asthma in early and late stages of the disease are necessary to elucidate the phenomena underlying the pathogenesis of these diseases and to develop technologies for targeted therapy and for compensation of the immune dysfunction. Therefore, the purpose of this study was to compare co-expression profiles of TNF receptors of types 1 and 2 in the main subpopulations of immunocompetent peripheral blood cells among RA patients or those with asthma with different disease durations and in conditionally healthy individuals in order to identify key trends within the course of these diseases and their prognostic and diagnostic markers.

## 2. Methods

To assess the levels of type 1 and 2 TNF receptor expression and co-expression, mononuclear cells were used that were isolated from peripheral blood of RA patients or those with asthma hospitalized in the Immunopathology Clinic at the Scientific Research Institute of Clinical Immunology (Novosibirsk, Russia). As a control group, data of conditionally healthy individuals (people without either RA or asthma) from the Novosibirsk Blood Center were used. The study was approved by the local ethics committee of RIFCI (protocol no. 24, dated 8 September 2016). All patients and healthy volunteers provided written informed consent for study participation and for publication.

Five study groups were compiled (Table 1) depending on the disease duration: RA patients with a disease duration of less than 10 years (group “early RA,” *n* = 26) or a disease duration of 10 years or more (group late RA, *n* = 15), patients with asthma with a disease duration of less than 10 years (group “early asthma,” *n* = 11) or 10 years or more (group late asthma, *n* = 11), and conditionally healthy individuals (the healthy group, *n* = 31).

The two groups of patients with early RA and asthma were comparable in sex (*p* = 0.442) and age (*p* = 0.682), and the same was true for the groups of patients with late RA and asthma (*p* > 0.999 and *p* = 0.443, respectively). Before admission to the hospital, patients with RA received glucocorticosteroids (3 (11.5%) patients in the early RA group and 1 (6.7%) patient in the late RA group), methotrexate alone or in combination with glucocorticosteroids (16 (61.5%) and 10 (66.7%) patients, respectively), leflunomide (4 (15.5%) and 2 (13.3%) patients, respectively) or sulfasalazine (3 (11.5%) and 2 (13.3%) patients, respectively), *p* > 0.999. All patients with asthma received beta-agonists and glucocorticosteroids before admission to the hospital; additionally, some patients received m-cholinoblockers (1 (9.1%) patient in the early asthma group and 2 (18.2%) patients in the late asthma group), leukotriene receptor antagonists (1 (9.1%) and 0 (0%), respectively), or their combination (1 (9.1%) and 4 (36.4%), respectively), *p* = 0.380.

Venous blood was collected from the cubital vein under sterile conditions in 6 mL vacuum tubes containing the K3-EDTA anticoagulant (tripotassium salt of ethylenediaminetetraacetic acid, Vacuette K3-EDTA, Greiner Bio-One GmbH, Kremsmünster, Austria) on an empty stomach.

Sample preparation was performed by means of lysis buffer (BD FACS Lysing Solution; cat. #349202; BD, Franklin Lakes, NJ, USA) according to the manufacturer’s instructions.

### 2.1. Flow Cytometry

Phenotypic characteristics of immune cells were assessed by flow cytometry (FACSVerse cytometer; BD, USA) using monoclonal antibodies: anti-CD25-FITC, anti-CD45R0-FITC, anti-CD19-PE-Cy7, anti-CD4-PE-Cy7, anti-CD3-PE-Cy7, anti-CD8-PE-Cy7, anti-CD127-PE-Cy7, anti-CD14-PerCP, and anti-CD45RA-Pacific Blue (Biolegend, USA); and anti-TNFR1-PE, anti-TNFR2-PE, anti-TNFR1-APC, and anti-TNFR2-APC (R&D Systems, USA). Data processing and calculation of fluorescence intensity parameters were performed in the FacsDiva software (BD, USA). To calculate the number of receptor molecules per cell, the BD QuantiBRITE PE Kit (BD Biosciences, USA) was used according to the method described earlier [22].

### 2.2. Statistics

Statistical analysis was carried out in STATISTICA 7.0 software (StatSoft, Tulsa, OK, USA). To determine the normality of the distribution in the studied parameters and the selection of statistical criteria, tests with the Shapiro–Wilk and Lilliefors (Kolmogorov–Smirnov) criteria were applied. The rejected data-normality-hypothesis-substantiated choice of non-parametric methods of statistical data analysis and presentation was selected for further analysis. Statistical significance for all tests was defined as *p* value < 0.05. The quantitative data are presented as a median (Me) and an interquartile range (IQR). A comparison of samples was performed using a nonparametric Mann–Whitney test (for comparison of two independent subgroups of patients) and through analysis of variance (Kruskal–Wallis rank test) with multiple comparison of medians (for comparison among 3 or more different subpopulations or studied subgroups of the human subjects). Associations between the studied parameters were evaluated using correlation analysis with Pearson’s coefficient.

## 3. Results

### 3.1. Co-Expression and Numbers of Receptor Molecules in the Main Populations of Mononuclear Cells

The co-expression and quantitative expression of TNF receptors of types 1 and 2 were assessed in the main immune-cell populations in RA and asthma patients (subdivided into groups according to the duration of the disease of either <10 or ≥10 years) and in healthy individuals (Figure 1).

In the comparison of groups, several significant trends were found in the changes in co-expression and quantitative expression of the receptors over the course of the diseases (depending on disease duration).

The most significant redistribution of receptor co-expression was noted in the B-lymphocyte population. All groups of patients differed significantly from healthy individuals in the percentage of double-negative cells. In the two asthma groups, this parameter was more than 4 times higher than that in healthy individuals (70.2% for early asthma and 62.3% for late asthma as compared to 15.9% for the healthy population, *p* = 0.023 and *p* < 0.001, respectively), whereas in the RA groups, this parameter was more than 2 times higher (43.6% for early RA (*p* = 0.183) and 35.4% for late (*p* = 0.005)). Moreover, after a long duration of the disease, a more than twofold increase in the proportion of TNFR1^+^ cells was observed among B cells, which was accompanied by a decrease in TNFR1 density in asthma and its increase in RA (*p* = 0.023 between groups late RA and late asthma).

RA patients, regardless of disease duration, were found to be characterized by a relatively stable proportion of TNFR2^+^ cells (as compared with healthy individuals) among monocytes, T lymphocytes, and regulatory T cells, with an increase in the proportion of double-positive cells among them, accompanied by a tendency for growing density of this receptor.

A distinguishing characteristic of early asthma proved to be a significantly higher proportion of TNFR1^+^ cells among (and a higher density of this receptor on) T cells in comparison with the healthy control and in comparison with RA patients. Additionally, only in early asthma was there a higher proportion of double-negative cells among regulatory T cells of up to 25%, which significantly exceeded this parameter in healthy individuals (5%, *p* = 0.002), late asthma (4.7%, *p* = 0.078), and early RA (7.3%, *p* = 0.047). For late asthma, the most characteristic feature was upregulation of the proportion of TNFR2^+^ cells to 98% among monocytes, accompanied by an insignificant increase in this receptor’s density.

### 3.2. Co-Expression and Numbers of Receptor Molecules in T Helper Cell Populations

The co-expression of the TNF receptors was assayed both in RA patients or asthma patients subdivided into groups according to disease duration and in healthy individuals on cells of T-helper subpopulations: CD4^+^, CD4^+^CD25^+^, CD4^+^CD45RA^+^, and CD4^+^CD45R0^+^ cells (Figure 2).

In terms of quantitative expression, the most significant difference was as follows: in all subpopulations of T-helper cells in the patients, regardless of disease duration, significantly lower TNFR1 density was observed as compared with healthy individuals. Meanwhile, activated T cells and memory T cells also manifested a higher proportion of TNFR1^+^ cells in asthma, starting from the early stage of the disease, but on the surface of naïve T-helper cells, these alterations began to appear only when the disease lasted more than 10 years.

Memory T-helper cells were characterized by the greatest differences in the co-expression of the two receptors. The group of healthy individuals differed in all types of expression from the groups of patients and exhibited the highest percentage of cells expressing the type 2 receptor (93.6%) and the lowest percentage of other cell subsets. In the patients, a lower proportion of TNFR1^+^TNFR2^−^ cells was observed. Both in RA and asthma patients, a sharply lower TNFR1 density was registered (more than 4.8-fold in asthma and more than 14.9-fold in RA, both *p* values<0.05). This shortage was accompanied by considerable heterogeneity of the density of the type 2 receptor on memory T-helper cells.

Characteristic aberrations in terms of naïve T-helper cells were seen both in early and late asthma. In early asthma, there was preservation of the co-expression profile seen in healthy individuals, along with an almost twofold lower density of the type 1 receptor (*p* < 0.0001). In late asthma, there was a much higher proportion of TNFR1^+^ cells as compared to early asthma, early RA, and healthy individuals.

A characteristic feature of RA patients was the absence of significant differences in the percentage of TNFR1/TNFR2-coexpressing subpopulations and in receptor density between early and late RA. On T-helper cells, all the anomalies characteristic of the early stage of RA persisted after 10 years of this disease.

### 3.3. Co-Expression and Numbers of Receptor Molecules in Cytotoxic Subpopulations of T Cells

We examined the TNF receptor co-expression on cells of cytotoxic subpopulations (CD8^+^, CD8^+^CD25^+^, CD8^+^CD45RA^+^, and CD8^+^CD45R0^+^) in both RA and asthma patients subdivided by disease duration and in healthy individuals (Figure 3).

Both the total pool and all three examined subsets of cytotoxic T lymphocytes showed a significant redistribution of the expression of TNFR1/R2 in patients compared with healthy individuals. The total pool, as well as activated cytotoxic T lymphocytes and memory cytotoxic T lymphocytes, proved to be characterized by the predominance of TNFR1^−^TNFR2^+^ cells (62.4%, 61.4%, and 78.8%, respectively), whose proportion was significantly lower in both RA and asthma patients than in the healthy controls.

A specific characteristic of patients with asthma was a significantly lower number of TNFR1 molecules on memory cells and naïve cells (both in early and late asthma) compared with healthy individuals. Meanwhile, the proportion of TNFR1^+^ cells, on the contrary, was significantly higher. Compared with healthy individuals, RA patients had a lower percentage of TNFR1^−^TNFR2^+^ cells among the activated cells and among the memory cells, as well as a higher percentage of TNFR1^−^TNFR2^+^ cells among naïve cytotoxic T lymphocytes.

The key difference between cytotoxic T cells among patients with early and late asthma was the combination of a dramatic increase in the proportion of double-positive cells with a significant decrease in TNFR2 number in late disease in both the total pool of CD8+ cells and in activated CD8+CD25+ cells.

Regardless of the state of health and duration of the disease, memory cytotoxic T lymphocytes were a subpopulation with nearly pervasive expression of at least one type of TNF receptor (94.1% of these cells in healthy individuals, 98.8% in early asthma, 100% in late asthma, 97.1% in early RA, and 94.3% in late RA).

### 3.4. Correlations of Expression and Co-Expression Parameters of TNF Receptors of Types 1 and 2 with the Duration of the Disease

We calculated these correlations as an additional assessment of the diagnostic and prognostic value of the identified differences between groups of patients with different disease durations (Table 2).

All statistically significant associations (*p* < 0.05) found among RA patients had low correlation coefficients (r = 0.33–0.42), denoting a trend rather than a significant association. Furthermore, changes in expression scores were substantially more strongly correlated with the patients’ age than with disease duration (eight vs. two statistically significant correlations, respectively). By contrast, for TNFR1/2 expression parameters in asthma, there was only one weak correlation with patients’ age (r = 0.466) while there were 24 correlations revealed with disease duration.

Another important feature of the identified correlations was the direction of the associations: longer disease duration was significantly correlated with an increase in the proportion of double-positive cells in asthma patients and with a decrease in the proportion of double-negative cells in them; the opposite pattern was documented in RA patients.

Just as in the other subpopulations, among cytotoxic T lymphocytes, we failed to detect specific significant differences in the co-expression and expression of TNF receptors that could differentiate between patients with early and late RA.

## 4. Discussion

Comparative evaluation of the expression and co-expression of TNF receptors in patients with two immune diseases differing in etiopathogenesis made it possible to detect an association of the presence and duration of the disease with a redistribution of the TNFR1–TNFR2 system and its adaptation to a long-term inflammatory process in the human body. In this analysis, distinctive distribution patterns of the type 1 and type 2 receptors on immunocompetent cells were identified that are characteristic of each disease in question and are of considerable basic-research and practical interest.

Aberrant TNF signaling mediated by the two types of receptors has been implicated in various immune-system-mediated diseases. The first important difference identified in the current study is the dissimilarity between the studied pathologies in correlations of TNFR expression parameters with disease duration. In asthma, a significant correlation of disease duration with the redistribution of receptors on the tested immune cells was demonstrated and was most pronounced for monocytes, naïve CD4^+^ cells, and cytotoxic T lymphocytes. Previously published papers also offer evidence that changes in TNF receptors’ expression reflect an altered function of various cell subtypes in this disease and likely reveal the predominance of different TNF signaling pathways in these cell subtypes [23,24,25]. In regulatory subpopulations, it is regarded as a functional marker of their suppressor activity, whereas in effector subpopulations, it serves as an activator of proliferation and cytotoxic capacity at an early stage of an immune response and as a signal of apoptosis at a late stage [26]. The oppositely directed changes in the expression profile of the type 1 and type 2 receptors that we identified in different subpopulations of cytotoxic T lymphocytes confirm this mechanism.

In contrast to asthma, in RA, the association of disease duration with the receptor expression data was much less significant. The key variable that was found to have a slightly more significant correlation with the redistribution of the TNF receptors is the age of the patients. Meanwhile, both demographic parameters were only weakly correlated with parameters of receptor expression. A search for diagnostic and prognostic biomarkers of RA and efficacy of therapy is underway [27]. Consequently, the absence of a significant effect of disease duration or patients’ age on the expression and co-expression of TNF receptors points to a crucial influence of other parameters, such as the type of RA course, ongoing therapy, the presence of local and systemic manifestations of the disease, and other variables associated with aggressiveness and intensity of the pathological process.

Recent research on the TNF signaling system and its complex interactions suggests that one of the mechanisms regulating cell sensitivity to cytokine action is expression and co-expression parameters of relevant receptors [28]. Elsewhere in cell models, we have demonstrated that one of the key co-expression parameters that determines the balance of proliferative and proapoptotic responses is the proportion of double-positive cells [8]. Moreover, other authors have reported a decrease in cell sensitivity to TNF in the absence of one of the receptors, which leads to serious disturbances in the functioning of the immune system and to the inability of the body to effectively combat various diseases [29,30]. In the current work, we also identified an important trend related to the proportion of double-positive cells: longer duration of the disease is significantly associated with the growth of the proportion of double-positive cells in asthma patients and a decline of this proportion in RA patients. This trend is suggestive of an alteration in the general sensitivity of cell populations to TNF and may reflect the magnitude of adaptive reactions of the human body to persistent inflammation. In this case, after prolonged asthma, the expansion of this population may be an attempt to compensate for the excessive action of TNF by switching proapoptotic responses to proliferative ones, in turn inducing pathological hyperreactive proliferation characteristic of late stages of asthma [31,32].

Despite the significant differences in TNF receptor expression profiles between RA and asthma, we noticed that five cell populations were characterized by the highest instability of TNF receptor expression in relation to the proportion of either double-negative or double-positive cells. The anomalies in these cells were well pronounced in both inflammatory diseases, and the parameters in question differed significantly from those of healthy individuals: CD19^+^ B cells (very few double-negative cells and many TNFR2^+^ cells with a high density of TNFR2), CD3^+^ T cells (the highest proportion of double-negative cells and the lowest density of TNFR2), and three populations with similar profiles of aberrations: regulatory T cells and activated CD4^+^CD25^+^ and CD4^+^ memory cells (the highest proportion of double-positive cells with the highest density of TNFR1). All these populations play an important role in the pathogenesis of both RA and asthma. In RA, B cells are the main cell type participating in both the initiation and progression of the disease. A release of proinflammatory cytokines, in particular TNF, promotes the infiltration of inflammatory cells into joints, thereby aggravating the formation of pannus and synovial hyperplasia and helping to maintain an aggregated B-cell infiltrate in the synovial membrane, the production of autoantibodies, and the formation of long-lived plasma cells, which are the main source of anti-citrullinated protein antibodies [22]. It has been reported that CD3^+^ T cells are found in most cases of early synovitis, and that the histological phenotype of synovial tissue samples is a predictor of disease persistence and severity; for instance, a lowered proportion of naïve CD4^+^ T cells is the strongest indicator of synovitis progression. Multivariate analyses of untreated and early-stage RA patients have confirmed that different phenotypes of CD4^+^ T cells can serve as a classifier distinguishing RA patients from healthy individuals [33]. There is evidence of a major role of T cells in asthma: namely, the balance among Th1, Th2, and Th17 cells. A release of cytokines by these cells contributes to the emergence of distinct signs of this disease, such as eosinophilia, mucus hypersecretion, and bronchial hyperreactivity [34]. B cells in asthma are the main producers of IgE, which is associated with persistent inflammation [35]. The regulatory-T-cell population is one of the key mechanisms controlling the development of inflammation in various diseases. Articles about RA and asthma suggest that a decline in the number of these cells is connected with the development and progression of these diseases [36,37].

In our comparison of the changes in TNFR co-expression profiles between the two pathologies, two key differences were found. First, overall, in the populations of immunocompetent cells of patients with asthma, there is a much higher percentage of cells expressing only the type 1 receptor, whereas in RA patients, the expression of the type 2 receptor is higher. These anomalies point to the importance of each type of receptor in the pathogenesis of the respective disease and are corroborated by multivariate regression models for predicting the onset and severity of the disease [38]. Secondly, in patients with asthma, most cell populations proved to be characterized by a unidirectional change in the relative and absolute number of receptor molecules correlating with disease duration; in other words, with an increase in the percentage of cells carrying the receptors, the density of the receptors on the cells went up too. In RA patients, oppositely directed changes were observed, i.e., with an increase in the percentage of cells carrying the receptors, a compensatory decrease in the density of the receptors was noted on these cells. This trend may explain the finding that during the atypical course of asthma and its resistance to standard glucocorticosteroid therapy, anti-TNF therapy [39,40] is sometimes effective and is typically used against RA specifically [41].

## 5. Conclusions

In our study, we revealed that there is a unique expression and co-expression profile of two types of TNF receptors that is associated with the pathogenesis of asthma and RA and therefore may become an additional diagnostic and/or prognostic criterion. Our results lay the foundation for subsequent evaluation of a functional response of an immune-cell population after alteration of the density of a TNF receptor and after a redistribution of the ratio of cells co-expressing the two TNF receptors. These data are important for a more complete understanding of regulatory mechanisms in a cytokine–receptor system. Future research is necessary to develop a therapeutic agent based on the co-expression profile of target cells, aimed at blocking or enhancing the expression of surface receptors to change local interaction with a cytokine instead of systemic anticytokine effects, which may lead to a decrease in the number of side effects of the therapy while maintaining its effectiveness.

## Figures and Tables

**Figure 1 cells-12-00005-f001:**
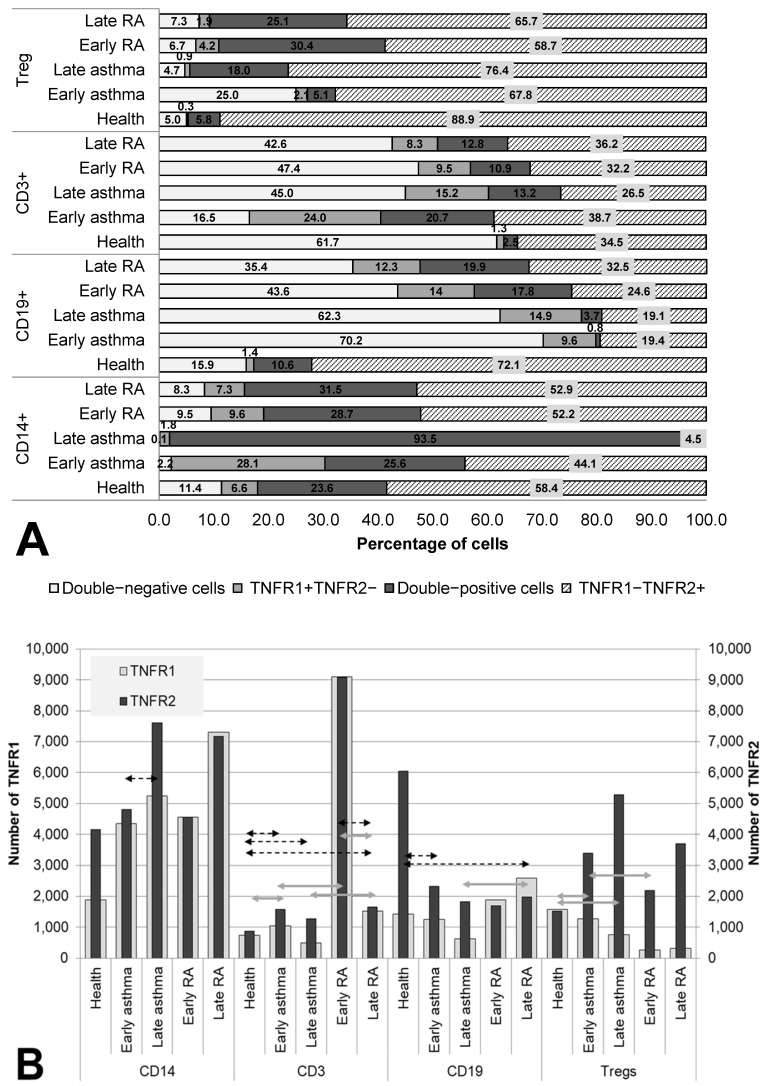
Co-expression (**A**) and density (**B**) of TNF receptors of types 1 and 2 in major subsets of immunocompetent cells. Arrows indicate significant (*p* < 0.05, Kruskal–Wallis test for multiple comparisons) differences in the expression level of TNFR1 between healthy individuals and RA patients or asthma and between patients at the same stage of the disease (between early RA and early asthma and between late RA and late asthma). Light grey arrows indicate differences for TNFR1, dark grey arrows indicate differences for TNFR2.

**Figure 2 cells-12-00005-f002:**
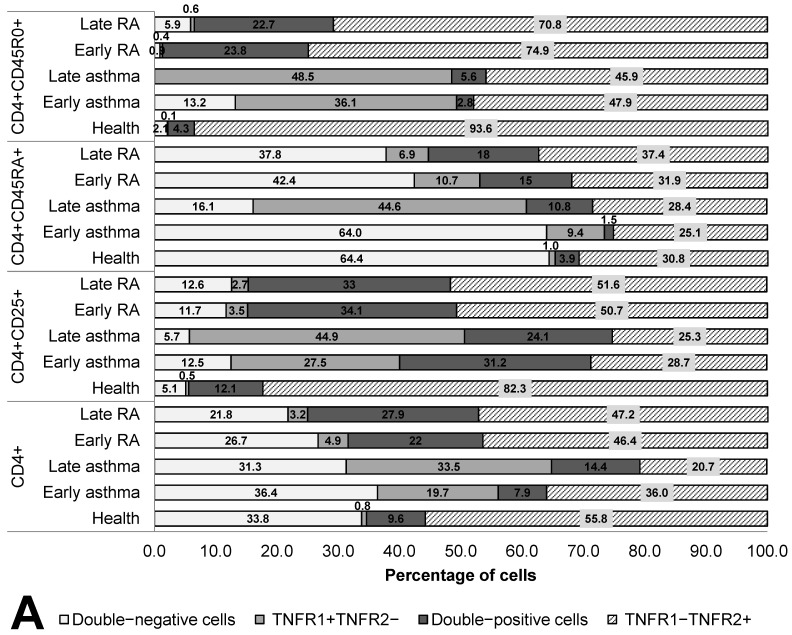
Co-expression (**A**) and density (**B**) of TNF receptors of types 1 and 2 in T helper cells subpopulations. Arrows indicate significant (*p* < 0.05, Kruskal–Wallis test for multiple comparisons) differences in the expression level of TNFR1 between healthy individuals and RA patients or asthma and between patients at the same stage of the disease (between early RA and early asthma and between late RA and late asthma). Light grey arrows indicate differences for TNFR1; dark grey arrows indicate differences for TNFR2.

**Figure 3 cells-12-00005-f003:**
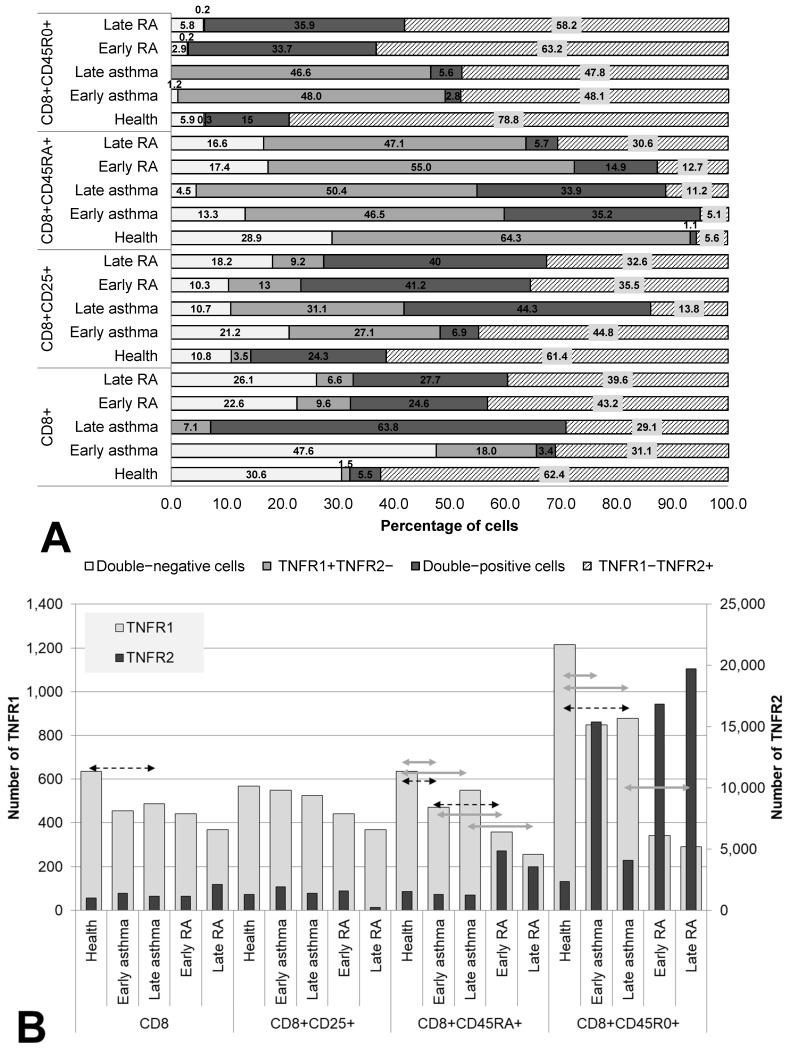
Co-expression (**A**) and density (**B**) of TNF receptors of types 1 and 2 in cytotoxic T lymphocytes subpopulations. Arrows indicate significant (*p* < 0.05, Kruskal–Wallis test for multiple comparisons) differences in the expression level of TNFR1 between healthy individuals and RA or asthma patients and between patients at the same stage of the disease (between early RA and early asthma and between late RA and late asthma). Light grey arrows indicate differences for TNFR1; dark grey arrows indicate differences for TNFR2.

**Table 1 cells-12-00005-t001:** Baseline characteristics of groups of human subjects.

Parameter	Disease Duration <10 Years	Disease Duration ≥10 Years	*p* Value
			(Mann–Whitney Test)
*RA patients*	Early RA group (*n* = 26)	Late RA group (*n* = 15)	
Sex: males, *n* (%)	6 (23.1%)	3 (20%)	>0.999
Age, median (IQR)	49.5 (33; 59)	48 (40; 61)	>0.999
Disease duration, median (IQR)	5 (4; 8)	14 (11; 17)	<0.001
Erosive arthritis, *n* (%)	19 (73.1%)	13 (86.7%)	0.445
Systemic manifestations of arthritis, *n* (%)	8 (30.8%)	9 (60%)	0.102
*Disease activity [n (%)]*			0/772
Low (DAS-28 < 3.2)	7 (26.9%)	3 (20%)	
Moderate (DAS-28 = 3.2–5.1)	10 (38.5%)	5 (33.3%)	
High (DAS-28 > 5.1)	9 (34.6%)	7 (46.7%)	
DAS-28, median (IQR)	4.47 (3.03; 5.43)	4.87 (3.81; 5.57)	
*Patients with asthma*	Early asthma group (*n* = 11)	Late asthma group (*n* = 11)	
Sex: males, *n* (%)	4 (36.4%)	2 (18.2%)	0.635
Age, median (IQR)	47 (42; 51)	46 (27; 57)	0.699
Disease duration, median (IQR)	3 (1; 6)	16 (12; 40)	<0.001
*Severity [n (%)]*			
Mild (FEV1 > 80%)	3 (27.3%)	1 (9.1%)	0.27
Moderate (FEV1 = 50–79%)	5 (45.5%)	4 (36.4%)	
Severe (FEV1 = 49–30%)	2 (18.2%)	6 (54.5%)	
FEV1, %: median (IQR)	70.05 (62.9; 81.1)	54.6 (46.5; 63.5)	
*Control group (healthy individuals)*	*n* = 30		
Sex: males, *n* (%)	9 (30%)		0.582 *
			>0.999 ^#^
Age, median (IQR)	45.5 (36; 58)		0.639 *
			0.876 ^#^

* as compared with RA patients, Kruskal–Wallis test, ^#^ as compared with asthma patients, Kruskal–Wallis test. Abbreviations. IQR: interquartile range, FEV1: forced expiratory volume in first second, DAS-28: Disease Activity Score for 28 joints.

**Table 2 cells-12-00005-t002:** Correlations between disease duration and patterns of expression of TNF receptors in immune-cell populations.

Parameter of Expression of TNFR1 and/or TNFR2	RA			Asthma		
	Patients’ Age	Disease Duration	Late RA	Patients’ Age	Disease Duration	Late Asthma
*CD14^+^ monocytes*						
% of double-positive cells					**0.718**	**0.869**
% of TNFR1^+^TNFR2^−^ cells					−0.483	−0.526
% of TNFR1^−^TNFR2^+^ cells					−0.562	−0.654
*n* * of TNFR1	0.397				0.546	0.519
*n* of TNFR2						0.537
*CD3^+^ T cells*						
% of TNFR1^−^TNFR2^+^ cells		0.329				
% of double-negative cells						0.564
*n* of TNFR1			−0.349		−0.499	−0.705
*n* of TNFR2					−0.497	−0.694
*regulatory T cells*						
% of double-positive cells	−0.342					
% of TNFR1^−^TNFR2^+^ cells	0.330					
% of double-negative cells					−0.541	−0.529
*n* of TNFR2						0.506
*CD4^+^ T helpers (total pool)*						
% of double-positive cells					0.522	
% of TNFR1^+^TNFR2^−^ cells						0.537
% of TNFR1^−^TNFR2^+^ cells					−0.619	−0.661
*n* of TNFR1						0.539
*CD4^+^CD25^+^ cells*						
% of double-negative cells						−0.580
% of double-positive cells						
% of TNFR1^+^TNFR2^−^ cells						0.578
% of TNFR1^−^TNFR2^+^ cells					−0.727	
*n* of TNFR2					0.729	0.756
*naïve CD4^+^ cells*						
% of double-negative cells					**−0.855**	**−0.862**
% of TNFR1^+^TNFR2^−^ cells					**0.821**	**0.868**
% of TNFR1^−^TNFR2^+^ cells					0.869	0.798
*n* of TNFR1						−0.588
*memory CD4^+^ cells*						
% of double-negative cells		0.393			−0.521	−0.667
% of double-positive cells	−0.338					
% of TNFR1^+^TNFR2^−^ cells						0.565
*n* of TNFR2					0.523	0.805
*CD8^+^ cytotoxic T cells (total pool)*						
% of double-negative cells					−0.588	**−0.849**
% of double-positive cells					0.555	**0.850**
% of TNFR1^+^TNFR2^−^ cells	−0.363					
% of TNFR1^−^TNFR2^+^ cells	0.361					
*CD8^+^CD25^+^*						
% of double-negative cells					−0.595	−0.518
% of double-positive cells					0.820	0.590
% of TNFR1^+^TNFR2^−^ cells	−0.420					
% of TNFR1^−^TNFR2^+^ cells					−0.741	−0.751
*n* of TNFR2					−0.685	−0.689
*naïve CD8^+^ cells*						
% of double-positive cells				0.466	0.567	
*n* of TNFR1					−0.489	
*memory CD8^+^ cells*						
% of TNFR1^−^TNFR2^+^ cells	0.343					

* Mean number of receptor molecules per cell. Pearson’s correlation coefficients (r) are presented with *p* value < 0.05. Bold highlights indicated strong correlations with r > 0.75.

## Data Availability

The data presented in this study are available on request from the corresponding author.

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
