# Peer review of "The Influence of Severity and Disease Duration on TNF Receptors’ Redistribution in Asthma and Rheumatoid Arthritis"

_cells, 2022, doi:10.3390/cells12010005_

Round 1
Reviewer 1 Report
The manuscript is of interest. Few points to consider:
Fig 2. The lower part should have Tcell subpopulations as shown in the upper part.
Table 2. What are the numbers: Pearson's correlation coefficient?
Paragraph (line 243-246 and Paragraphe line 263: No clear meaning. Please rephrase
Ref list. Few refs not complete. The volume and/or pages of the journal are missing
Author Response
Fig 2. The lower part should have Tcell subpopulations as shown in the upper part.
Answer: Part of the drawing has shifted and covered the captions; it was fixed.
Table 2. What are the numbers: Pearson's correlation coefficient?
Answer: That's right, an explanation was added to the table legend: the table shows Pearson's correlation coefficients, only those with p values lower than 0.05.
Paragraph (line 243-246 and Paragraphe line 263: No clear meaning. Please rephrase
Answer: Sentences were rephrased to be more accurate.
Ref list. Few refs not complete. The volume and/or pages of the journal are missing
Answer: References №№2,7,11,21,22,23,25,26,30 and 35 were completed.
Reviewer 2 Report
The authors of the present study aim to evaluate the redistribution patterns of cytokine receptors, specifically type 1 and 2 receptors of tumor necrosis factor. This evaluation was performed on peripheral blood cells in two groups of patients, patients with rheumatoid arthritis or patients with asthma. These groups were stratified according to disease duration.
At first reading the title the idea is that the study was conducted longitudinally and prospectively as they mention that the redistribution is over the course of asthma and rheumatoid arthritis. However, I believe that according to their initial objective this is not like that.
I describe below some points to review
1. There are typographical errors in the figures as some of them read Early BA but never mention what BA means or whether it is actually RA.
2. It should be considered that for statistical analyses and multiple comparisons in the results shown in figures 1, 2 and 3, for comparisons between the two TNF receptors for each cell subtype and between the different groups of patients with RA or asthma the most appropriate statistical analysis to perform should be a two-way ANOVA. This analysis allows to decide whether there is a significant interaction between patient groups and the proportion of each of the TNF receptors. In addition, it would avoid statistical errors and give more certainty to the results described.
3. The authors made a graph with the percentage of cells for each cell subtype and between the different groups of patients apparently for figures 1, 2 and 3, however these seem to be in fact different figures as there is no further description of them in the figure captions so they should be as independent figures, and perhaps consider placing them as supplementary material as I consider that they do not provide more relevance than the figures where the comparison between receptors, patients and cell subtypes is made. There is no statistical analysis of proportions in these figures.
4. Several of the graphs have letter splices and it is difficult to read the text in the figures properly.
5. In the description of material and methods in the statistical analysis section the authors mention the analysis for comparison between two groups and a kruskal-Wallis analysis for multiple comparisons. However, one has to assume that the comparison where the Mann-Whitney U-test is used is the one for demographic data, but they never mention it in the table. Furthermore, they never mention the statistical test to assess the normality of the data and only assume the use of two non-parametric tests without mentioning this normality test.
6. Similarly to the previous point, in material and methods in the statistical analysis section the authors never mention the statistical tests they used to assess the correlation of the variables mentioned in table 2, which again leaves it to the reader's discretion to assume that the correlations presented have a significant p but they never report it in the table, only mentioning that they are less than 0.05. It is also not mentioned why some numbers are highlighted in bold and others are not.
7. In the title and objective of the paper the authors imply that the differences between the proportions of TNF receptors are given between the duration and severity of the disease for RA or asthma, however what is observed in the results, in the discussion and in figures 1, 2 and 3, is rather a comparison between diseases where the proportions of TNF receptors in early or late stage are compared but between diseases, so that the comparisons seem to be discordant to the title and objectives.
Author Response
At first reading the title the idea is that the study was conducted longitudinally and prospectively as they mention that the redistribution is over the course of asthma and rheumatoid arthritis. However, I believe that according to their initial objective this is not like that.
Answer: Thank you for this clarification, we have rephrased the title of the article to be more accurate in conveying the purpose of the study.
I describe below some points to review
- There are typographical errors in the figures as some of them read Early BA but never mention what BA means or whether it is actually RA.
Answer: The legends on the figure 3 have been changed.
- It should be considered that for statistical analyses and multiple comparisons in the results shown in figures 1, 2 and 3, for comparisons between the two TNF receptors for each cell subtype and between the different groups of patients with RA or asthma the most appropriate statistical analysis to perform should be a two-way ANOVA. This analysis allows to decide whether there is a significant interaction between patient groups and the proportion of each of the TNF receptors. In addition, it would avoid statistical errors and give more certainty to the results described.
Answer: Thanks for the revealed inaccuracy in the description of the statistical criteria in the captions to the figures, the description was incomplete. We added description in the legends for all figures and in the Methods section. Since matrix multiple comparisons were required when comparing the scores of patients with asthma and RA with those of healthy donors (3 comparisons for each phase of the disease and 4 for healthy donors), statistical criteria adjusted for multiple comparisons are needed here. To do this, we chose the nonparametric rank dispersion Kruskal-Wallis test, which in this situation produces more rigorous and accurate values compared to ANOVA due to the additional correction for multiple comparisons.
- The authors made a graph with the percentage of cells for each cell subtype and between the different groups of patients apparently for figures 1, 2 and 3, however these seem to be in fact different figures as there is no further description of them in the figure captions so they should be as independent figures, and perhaps consider placing them as supplementary material as I consider that they do not provide more relevance than the figures where the comparison between receptors, patients and cell subtypes is made. There is no statistical analysis of proportions in these figures.
Answer: We agree with the need to more clearly indicate in the description of the figures what information is presented in them. The legends of figures 1,2 and 3 have been expanded and modified for greater accuracy.
Regarding the transfer of the upper part of the figures from the main text to the supplementary material due to the lack of additional information in them. We do not agree with this, as the lower part of the figures can not provide information about the proportion of co-expressing cells (eg, the proportion of double-positive cells), it only provide information about number of receptors per cell. Both percentage of cells and receptors density were demonstrated in our study as important parameters associated with the severity and duration of the disease. Comparison of all 4 fractions of co-expressing cells is an important part of the study and should be placed in the main part of the article.
Regarding the statistical analysis of proportions. Since the presented numbers do not reflect the average proportions of integer values (for example, patients), but they reflect the mean proportions of percentage data (from hundreds of thousands of cells analyzed on a flow cytometer for each person), no tests for proportions (Fischer's exact test, chi-square, and others) are not applicable for them. At the same time, statistical differences in the percentage of individual cell fractions (for example, TNFR1+TNFR2-) between patient subgroups were determined for each cell type using the Kruskal-Wallis test. It is not possible to arrange all the obtained values in the figures due to the cumulative nature of the diagram, therefore the most significant results with an indication of the level n are given in the text under the figures.
- Several of the graphs have letter splices and it is difficult to read the text in the figures properly.
Answer: We have modified drawings and legends for a better perception of information.
- In the description of material and methods in the statistical analysis section the authors mention the analysis for comparison between two groups and a kruskal-Wallis analysis for multiple comparisons. However, one has to assume that the comparison where the Mann-Whitney U-test is used is the one for demographic data, but they never mention it in the table. Furthermore, they never mention the statistical test to assess the normality of the data and only assume the use of two non-parametric tests without mentioning this normality test.
Answer: Thank you for the important clarification. To determine the normality of the distribution in the studied parameters and the selection of statistical criteria, tests with the Shapiro-Wilk and Lilliefors (Kolmogorov-Smirnov) criteria were applied. The data were considered non-normally distributed if at least one test failed to confirm the hypothesis of normality. As a result, all data were determined to be non-normally distributed, and non-parametric criteria were chosen. This information has been added to the Methods section.
- Similarly to the previous point, in material and methods in the statistical analysis section the authors never mention the statistical tests they used to assess the correlation of the variables mentioned in table 2, which again leaves it to the reader's discretion to assume that the correlations presented have a significant p but they never report it in the table, only mentioning that they are less than 0.05. It is also not mentioned why some numbers are highlighted in bold and others are not.
Answer: For each correlation coefficient given in the article, the significance p value was automatically calculated. Table 2 shows only those correlations for which the p value lower than 0.05. An addition of individual p values to each of r coefficient in table would greatly complicate the perception of the table. We have changed the description of correlation analysis in Methods and in the table legend.
- In the title and objective of the paper the authors imply that the differences between the proportions of TNF receptors are given between the duration and severity of the disease for RA or asthma, however what is observed in the results, in the discussion and in figures 1, 2 and 3, is rather a comparison between diseases where the proportions of TNF receptors in early or late stage are compared but between diseases, so that the comparisons seem to be discordant to the title and objectives.
Answer: Thanks for the valuable clarification. In our study, the main goal was to compare the parameters of patients with the same disease at its early and late stages. However, after adjusting for a statistical adjustment for multiple comparisons (due to 2+ comparisons with a control group), only a small number of significant differences were found directly between the early and late RA or early and late asthma groups. Therefore, in addition, we conducted several more types of comparisons, and the final manuscript included (and it is shown in the figures) 4 types of comparisons:
1) between the early and late phase within each disease;
2) between each phase of each disease and indicators of healthy donors;
3) between early arthritis and early asthma;
4) between late arthritis and late asthma.
We also modified the figures in order to make the key differences more visible and relevant to the goals of the study.
Reviewer 3 Report
Manuscript ID: Cells-1949548
“The influence of severity and long disease duration on TNF receptors’ redistribution in the course of asthma and rheumatoid arthritis”
Comments to the Authors:
In this study, the authors describe the differences in the expression and co-expression profiles of TNF receptor types 1 and 2 in immunocompetent cell populations in patients with RA or asthma, with the aim of comparing the co-expression profiles of TNF receptors with different disease durations. These results would hope to improve the prognostic and diagnostic markers of diseases.
The topic is interesting and the specific approach is suggestive. The experiments were well conducted, however, the conclusions of the study are not so definitive and useful.
There are some specific questions:
- The expression of these receptors was also analyzed in other autoimmune or allergic diseases?
- The authors split the AR patients in two groups based on disease duration, there are data on early patients naïve to any treatment?
- In the description of characteristics of patients the authors should add data on the therapy followed by the patients.
- Patients with Rheumatoid Arthritis used anti-TNFa drugs? In any case, there are indications about the expression of the receptors on the cells of patients subjected to this therapeutic treatment?
- In conclusion, the results of this work what do they suggest about the evaluation of these receptors? What clinical indications can be provided?
- In the figures it would perhaps be clearer to use the symbols instead of the arrows to indicate the statistical significance, or in any case describe them better in the legends.
Author Response
The topic is interesting and the specific approach is suggestive. The experiments were well conducted, however, the conclusions of the study are not so definitive and useful.
Answer: Thank you for your valuable comment. We have rephrased the conclusions to clarify the possibility of further use of our results.
There are some specific questions:
- The expression of these receptors was also analyzed in other autoimmune or allergic diseases?
Answer: At this stage of the study, we selected only these two diseases from a larger spectrum previously analyzed with a smaller number of parameters (the preliminary experiments also included atopic dermatitis and tuberculosis, as representatives of other immunological disorders). At the moment we are developing an application for a local patent for a method for early diagnosis of diseases using our technology. After its completion, we plan to expand our work with other cytokines and other pathologies.
- The authors split the AR patients in two groups based on disease duration, there are data on early patients naïve to any treatment?
Answer: Unfortunately, the hospital of our clinic does not admit patients with a newly established diagnosis and naïve in terms of therapy, which did not give us the opportunity to analyze such an interesting subgroup.
- In the description of characteristics of patients the authors should add data on the therapy followed by the patients.
Answer: Information on prior therapy has been added to the Methods section
- Patients with Rheumatoid Arthritis used anti-TNFa drugs? In any case, there are indications about the expression of the receptors on the cells of patients subjected to this therapeutic treatment?
Answer: Not receiving anti-TNF therapy was one of the conditions for enrolling patients in our study, since at the current stage it was important for us to determine changes in the TNF system without exposure to targeted drugs. Only after determining the standard expression profile characteristic of this pathology, it seems possible to further study its changes under the influence of anti-TNF drugs and evaluate the significance of these changes.
- In conclusion, the results of this work what do they suggest about the evaluation of these receptors? What clinical indications can be provided?
Answer: Further development of our work is associated with mathematical modeling and the construction of complex multifactorial prognostic and diagnostic models of response to therapy based on the patterns of receptor redistribution parameters that we have identified at this stage, as well as with the search for new therapeutic targets (blocking local receptor expression instead of systemic anti-cytokine effects). The conclusion of the study was reformulated to accommodate these assumptions.
- In the figures it would perhaps be clearer to use the symbols instead of the arrows to indicate the statistical significance, or in any case describe them better in the legends.
Answer: Thanks for the valuable clarification, we have changed the legends for the figures to more accurately understand the colors of the arrows. We have also modified the drawings themselves for better perception.
Round 2
Reviewer 3 Report
Comments to the Authors:
The Authors with their comments have fully satisfied my requests.